# Role of Healthcare Professionals and Sociodemographic Characteristics in COVID-19 Vaccination Acceptance among Uro-Oncology Patients: A Cross-Sectional Observational Study

**DOI:** 10.3390/vaccines11050911

**Published:** 2023-04-28

**Authors:** Predrag Nikic, Branko Stankovic, Veljko Santric, Ivan Vukovic, Uros Babic, Milan Radovanovic, Nebojsa Bojanic, Miodrag Acimovic, Luka Kovacevic, Nebojsa Prijovic

**Affiliations:** 1Clinic of Urology, University Clinical Center of Serbia, 11000 Belgrade, Serbia; nicksha@gmail.com (P.N.); bstank@gmail.com (B.S.); veljkosantric@yahoo.com (V.S.); ivanvukovic.urolog@gmail.com (I.V.); urosb2001@yahoo.com (U.B.); milan_950@hotmail.com (M.R.); bojanicnebojsa@gmail.com (N.B.); drmaiodragacimovic@gmail.com (M.A.); 2Faculty of Medicine, University of Belgrade, 11000 Belgrade, Serbia

**Keywords:** COVID-19, vaccination, acceptance, healthcare professionals, sociodemographic characteristics

## Abstract

At the time when mass COVID-19 vaccination began, providing appropriate vaccination advice to uro-oncology patients became a challenge. This was a single-center cross-sectional observational study aimed to investigate the rate of COVID-19 vaccination among uro-oncology patients receiving systemic therapy for metastatic renal cell carcinoma and metastatic castration-resistant prostate cancer. Furthermore, we aimed to assess patients’ attitudes and identify factors influencing their decision to vaccinate against COVID-19. Data on patients’ sociodemographic characteristics, vaccination status, and awareness and attitudes about COVID-19 vaccination were collected from questionnaires completed by the patients. A total of 173 patients were enrolled in this study, and 124 (71.7%) of them completed the COVID-19 vaccination. Significantly higher vaccination rates were found in male patients, and also in older patients, highly educated patients, and those who lived with one household member. Furthermore, we found significantly higher vaccination rates in patients who had consulted with doctors involved in their treatment, particularly with urologists. A significant association was observed between COVID-19 vaccination and doctor’s advice, family member influence, and personal beliefs toward the vaccination. Our study showed multiple associations of patients’ sociodemographic characteristics with vaccination rates. Furthermore, consultation with doctors who are particularly involved in oncology treatment and advice received from them were associated with significantly higher vaccination rates among uro-oncology patients.

## 1. Introduction

Soon after the first case of SARS-CoV-2 infection was reported in Wuhan, China, in December 2019, the COVID-19 pandemic rapidly spread, causing significant changes in all areas of life across the globe [1]. From the beginning of the pandemic until November 2022, there have been over 630 million cases of COVID-19 and approximately 6.5 million deaths worldwide [2]. Although in many cases the clinical presentation was mild, risk factors for severe clinical outcomes and potentially fatal consequences of COVID-19 infection have been identified. In a systematic review and meta-analysis conducted by Booth et al., oncology patients were recognized as a risk group for worse COVID-19 outcomes. Along with active malignant disease, age over 75, male gender, and associated comorbidities were all recognized as risk factors for severe COVID-19 disease [3]. The introduction of vaccines against COVID-19 has provided a glimmer of hope in the long and exhausting fight against this disease on all fronts. Upon recognizing that malignant disease is a risk factor for the development of more severe forms of COVID-19 and potentially fatal outcomes, the immunization of oncology patients became a priority [4]. However, at the time when mass vaccination began, sufficient data on vaccine efficacy for oncology patients were lacking [5]. Soon after, the European Society for Medical Oncology (ESMO) advocated for the vaccination of oncology patients, but still, the existence of many uncertainties about the efficacy and safety of vaccination in this patient population remained [6].

Therefore, it was important to understand the impact of specific oncology treatments on patients’ immune systems and to reproduce the experiences with vaccines against other infectious diseases to provide appropriate vaccination guidance [7]. In Serbia, medical treatment for uro-oncology patients has been continuously provided since the beginning of the pandemic, despite numerous health system reorganizations and movement restrictions during lockdowns. Systemic targeted treatment for metastatic renal cell carcinoma (mRCC) and metastatic castration-resistant prostate cancer (mCRPC) in Serbia is only available in a few healthcare centers, including the Clinic of Urology, University Clinical Center of Serbia.

From the beginning of 2021, vaccines against COVID-19 from four different manufacturers were widely available in Serbia, including mRNA, vector, and inactivated vaccines, with the option to choose a vaccine free of charge [8]. However, this opportunity to choose a vaccine has led to ambiguity and potential non-medical influences on the population’s decisions toward vaccination [9]. Thus, providing appropriate vaccination advice to uro-oncology patients became a challenge for doctors involved in the treatment of this patient population. Due to the relatively short period of widespread vaccine availability, data on vaccination and vaccination attitudes among uro-oncology patients, especially those receiving systemic targeted therapy, are limited.

This study aimed to investigate the rate of COVID-19 vaccination among uro-oncology patients receiving systemic targeted therapy with tyrosine kinase receptor inhibitors (TKIs) for mRCC and systemic therapy with androgen receptor target agents (ARTAs) for mCRPC. Furthermore, we aimed to assess patients’ attitudes and identify factors influencing their decision to vaccinate against COVID-19.

## 2. Materials and Methods

This was a single-center cross-sectional observational study that enrolled 173 patients who were receiving systemic targeted therapy with tyrosine kinase receptor inhibitors (TKIs) for mRCC and androgen receptor target agent (ARTA) therapy for mCRPC. The study was conducted at the Clinic of Urology, University Clinical Center of Serbia (UKCS) in March 2022, one year after COVID-19 vaccines became widely available in Serbia. All patients who were currently receiving therapy and had completed at least one cycle of systemic treatment were eligible for the study. All patients who agreed to participate provided written consent. Data on clinical characteristics, underlying oncological disease, and systemic therapy were obtained from patients’ electronic and paper medical records. We collected information on patients’ sociodemographic characteristics, vaccination status, awareness and attitudes about COVID-19 vaccination, and factors that influenced their decisions from questionnaires completed by the patients. This study was approved by the Ethics Committee of the University Clinical Center of Serbia.

### 2.1. Questionnaire

A structured questionnaire was developed for this study at the Clinic of Urology, UKCS, by consensus among all authors. The questionnaire used in this study consisted of 21 closed-ended questions with options for choosing one or more answers. The first part of the questionnaire covered sociodemographic characteristics, the presence of comorbidities, and the current vaccination status against COVID-19. The sociodemographic characteristics included gender, age, permanent residence, level of education, whether the patient or their household members had a medical education, and the number of family members with whom the patient lives. If existing, significant comorbidities such as cardiovascular diseases, diabetes mellitus, obesity, chronic obstructive pulmonary disease, and chronic renal failure were recorded for each patient. The current vaccination status was determined by asking whether the patient had received at least 2 doses of the COVID-19 vaccine. In the second part of the questionnaire, patients answered how they were informed about the COVID-19 vaccine and which factors most influenced their decision to or not to vaccinate. All patients were also asked about previous COVID-19 infection and the date of the infection. Vaccinated patients answered questions about hesitancy when deciding to vaccinate, the number of doses they received, and the dates of vaccinations. Unvaccinated patients were asked about a potential change in attitude and decision to vaccinate at that moment. The questionnaire was administered to patients during routine clinical visits at the Clinic of Urology, UKCS. The patients were informed about the purpose and importance of the study and were allowed to ask questions before giving their consent to participate in the research. The questionnaires were self-administered, and patients could ask for assistance from the study staff if needed. The full version of the questionnaire is available in Appendix A.

### 2.2. Statistical Analysis

The statistical analysis of data was conducted using descriptive and analytical statistics. Student’s *t*-test was used to analyze the significance of the difference for variables with a normal distribution between patient groups, while the Mann–Whitney U-test was used for variables that did not have a normal distribution. Differences in frequency among patient subgroups were analyzed using the χ^2^ test or Fisher’s exact probability test. A *p*-value of less than 0.05 was considered statistically significant. The statistical software package SPSS version 20 for Windows was used for data processing.

## 3. Results

This single-center cross-sectional observational study included 173 patients who were receiving systemic targeted therapy with TKIs for mRCC and ARTA therapy for mCRPC. The sociodemographic and clinical characteristics of the study group are shown in Table 1. The mRCC group encompassed 93 patients (53.8%) and 80 patients (46.2%) were in the mCRPC group. Most of the patients included in the study were men (*n* = 149, 86.1%), while only 24 (13.9%) were women in the mRCC group. The median duration of the underlying malignant disease was 51 months, and the median duration of systemic treatment was 11 months. In the answers to the questions, 103 (59.5%) patients reported living outside the capital, 25 (14.5%) patients reported living alone, 71 (41.0%) reported living with one family member, and 77 (44.5%) reported living with two or more household members. In terms of education level, 97 (56.1%) patients had high school, and 49 (28.3%) had higher education. Only 3 (1.7%) patients were medical professionals, while 32 (21.6%) reported that they had medical professionals in their household. Accompanying comorbidities were found in the majority of patients (*n* = 111, 64.2%), with cardiovascular diseases being the most prevalent (61.3%).

Table 2 summarizes the vaccination status and patients’ attitudes related to vaccination against COVID-19. In the entire studied population, 124 (71.7%) patients completed vaccination against COVID-19, without hesitancy to receive the vaccine reported in about 90% of cases. The majority of participants (*n* = 128, 74%) reported that they were sufficiently informed about vaccination against COVID-19. Before vaccination, 127 (73.4%) patients consulted with their doctors about vaccination, the most often with a urologist (47.4%) and/or chosen primary care physician (41.0%). The majority of patients who consulted with doctors about vaccination (84.3%) were advised to complete the vaccination. As shown in Table 2, more than two-thirds of patients living with at least one household member (*n* = 106, 71.6%) had a family member vaccinated against COVID-19 as well. Among the factors that influenced the decision to receive the vaccine, the most often, patients reported doctor’s advice (*n* = 64, 37.0%), underlying malignant disease (*n* = 63, 36.4%), and personal beliefs (*n* = 61, 35.3%). In the unvaccinated group, only seven (14.9%) patients acknowledged a change in attitude towards vaccination against COVID-19; i.e., they were ready to complete vaccination at that moment. At the moment of the survey, 108 (62.4%) patients reported that they had not contracted COVID-19 infection ever before.

Table 3 presents the association between the clinical and sociodemographic characteristics of patients and actual vaccination status. Our analysis showed that vaccinated patients were significantly older than unvaccinated patients, with a mean age of 68.6 ± 9.4 compared to 64.1 ± 9.5 (*p* = 0.006), respectively. Furthermore, a significantly higher proportion of men were vaccinated compared to female patients, with vaccination rates of 75.8% and 45.8%, respectively (*p* = 0.002). In terms of education level, almost 90% of highly educated patients had received the COVID-19 vaccine, while the vaccination rates for patients with high school education and those with elementary school education or less were 67.0% and 55.6%, respectively (*p* = 0.002). Furthermore, we found a significant association between the number of household members and vaccination status. To be specific, patients who lived with one household member had a higher vaccination rate of 81.7%, while a lower frequency of vaccination was observed among patients who lived alone (68.0%), but also in those who lived with two or more family members (63.6%) (*p* = 0.047). When analyzing several clinical and sociodemographic characteristics such as underlying malignant disease, duration of malignant disease, duration of systemic treatment, permanent residence, occupation, family members with medical education, and presence of comorbidities, we found no significant differences between the vaccinated and unvaccinated patient groups.

Table 4 displays the association between the information received and attitudes toward vaccination in the vaccinated and unvaccinated groups of patients. Our findings indicate that patients who had consulted with doctors involved in their treatment were after that vaccinated at a significantly higher rate than those who had not (79.5% vs. 50.0%, respectively, *p* < 0.001). According to doctors’ specialty, a significantly higher number of patients who consulted their urologist were vaccinated (89%, *p* < 0.001). Among patients who received doctors’ advice to vaccinate, 87.9% of them received the COVID-19 vaccine, while only 16.7% of patients who received doctors’ advice against vaccination decided to complete vaccination (*p* < 0.001). When analyzing factors influencing the decision to receive the vaccine, a significant association was observed between COVID-19 vaccination and doctor’s advice (92.2% were vaccinated, *p* < 0.001), family member influence (95.2% were vaccinated, *p* = 0.011), and personal beliefs towards the vaccination (60.7% were vaccinated, *p* = 0.018). Patients who did not have a history of COVID-19 infection prior to the survey were vaccinated significantly more often (77.8%) compared to patients who contracted COVID-19 before the start of systemic therapy (48.0%) or during treatment (70.0%) (*p* = 0.011).

## 4. Discussion

In this study, we investigated factors influencing the decision to vaccinate against COVID-19 in a specific population of patients who were receiving systemic targeted therapy with TKIs for mRCC and ARTA therapy for mCRPC, at a time when there was no clear evidence regarding the vaccination for this group of patients against COVID-19. Compared to patients with urological malignancies who receive systemic therapy administered in a hospital setting, such as immunotherapy and chemotherapy, TKI and ARTA therapies are both administered orally and do not require hospitalization. Keeping in mind that patients who received systemic therapy in hospital conditions were exposed to the risk of both possible treatment interruptions and restrictions related to limited hospital admissions during the pandemic, patients who received oral systemic oncology therapy had a significant advantage and comfort.

Our results showed that 71.7% of participants had received the COVID-19 vaccine, which was significantly higher when compared to the vaccination rate of 48% in the general population in Serbia at the same time [10]. In contrast to our findings, a study by Bain et al. did not show a significant difference in the vaccination rate between oncology patients and the general population in Australia, although the overall vaccination rate in that country was higher [11]. However, another study conducted in one of the largest oncology centers in Serbia in 2021, which included patients with both solid and hematological malignancies, reported the highest vaccination rate in patients with genitourinary tumors [12]. We can explain the difference in vaccination rates between our respondents and the general population in Serbia based on at least three factors. First, given that malignant disease was identified as a risk factor for severe COVID-19 infection early in the pandemic, oncology patients were prioritized for vaccination [4]. Second, the study by Matovina Brko et al. suggests that patients with genitourinary tumors in Serbia were more motivated to receive vaccination than other oncology patients [12]. Finally, Serbia had a relatively low COVID-19 vaccination rate in the general population compared to other countries at the time of our study [10].

Furthermore, when analyzing the sociodemographic characteristics of our patients, we found that the vaccinated group was significantly older than the unvaccinated group of patients. Our results are consistent with previously published studies that have shown a higher vaccination rate among older people, including oncology patients [11] and other populations [13]. This finding is not surprising when knowing that the older population was considered at high risk for severe outcomes of COVID-19 infection, making immunization for this population essential [3]. Additionally, our results showed that the vaccination rate was significantly higher in male patients. Similar to our results, a study conducted on a population that had high compliance rates with common vaccines stressed that the female gender is one of the factors for not accepting the vaccine against COVID-19 [14]. Moreover, the meta-analysis by Booth et al. showed that men are at greater risk for more severe forms and outcomes of COVID-19 infection [3], which can potentially explain our findings. However, in contrast to our results, other studies have shown higher acceptance of vaccination among women [13,15], but notably in samples predominantly consisting of women participants. Conflicting results obtained in our study can be partially attributed to the gender imbalance with predominantly male patients included in the study as well (86.1%). In further analysis of sociodemographic characteristics, we found that highly educated patients had the highest vaccination rate. Several studies have underlined the association of a lower level of education with negative attitudes towards vaccination [14,15]. Moreover, a study among healthcare workers found that doctors were vaccinated more often than nurses against COVID-19 [13].

Based on our results, we found a significant influence of family factors on patients’ decision to vaccinate. Namely, patients who lived with one household member had the highest vaccination rate compared to those who lived alone or with more than one household member. Our findings also showed that patients whose family members were vaccinated against COVID-19 and those who reported that influence by a family member was important in the decision to complete vaccination were more likely to be vaccinated themselves. Comparable to our findings, pre-pandemic research in four European countries indicated a significant influence of family and/or friends on the decision to vaccinate against respiratory diseases [16]. It is not surprising that we noticed a higher vaccination rate in patients who have not contracted COVID-19 previously, which is consistent with findings from other studies showing higher vaccine acceptance among patients who had a stronger fear of COVID-19 infection [17].

The importance of doctors in promoting vaccine acceptance among patients is well known. Due to the uncertainties surrounding the appearance of vaccines against COVID-19, healthcare professionals were recognized as critical sources in providing information about the vaccine. Previous studies in pediatric populations have confirmed the importance of strong vaccination recommendations obtained from healthcare professionals [18,19,20]. Our study supports these findings, showing that patients who consulted with doctors involved in their oncology treatment were more likely to be vaccinated against COVID-19. It is noteworthy that healthcare professionals may also have concerns about vaccines, particularly when it comes to new vaccines [21]. Vaccine hesitancy among healthcare workers can influence the general population’s attitude towards vaccination [22]. Therefore, it is essential that healthcare workers are well informed about vaccines and prepared to discuss vaccination with patients, who may have access to contradictory information about vaccines in media and the internet [23]. Our results confirm the importance of doctors in promoting vaccination. Specifically, a higher vaccination rate (87.9%) was found in patients who were advised to vaccinate by their doctors. On the other hand, the majority of patients advised against vaccination by their doctors (83.3%) were not vaccinated at the time of this study. Furthermore, our findings show that the doctor’s advice alone was a statistically significant factor influencing patients’ decisions to vaccinate. Obtained results are in line with a study conducted among oncology patients in Cyprus, which found that patients who had received information about COVID-19 vaccines from healthcare professionals were more willing to accept the vaccine [24]. Our research indicates that patients predominantly followed their doctors’ advice, emphasizing the importance of the doctors’ recommendations in the patient’s decision to vaccinate. Studies conducted before the COVID-19 pandemic demonstrate that effective communication between healthcare professionals and patients can reshape even the most skeptical patients’ opinions on vaccination [25,26].

Considering that our study population consisted of patients with urological malignancies who were receiving systemic treatment, we have examined the role of urologists in promoting vaccination with special interest. We have found that patients who consulted with their urologist about COVID-19 vaccination were more likely to complete vaccination (89.0%) compared to those who consulted with doctors of other specialties. Furthermore, research by Marijanović et al. signified the importance of the chosen primary care physician in promoting COVID-19 acceptance vaccine among oncology patients [27], but this study included patients with multiple cancers and did not examine the influence of physicians involved in specific oncology treatment. However, our findings emphasize the significant influence that prescribing urologists have on the decision of uro-oncology patients to vaccinate. Both TKI and ARTA treatments that the patients were receiving in our study represent lifelong therapy. Hence, the interpretation of our findings is supported by previously published research which showed that patients’ trust in oncologists is built not only on doctors’ medical knowledge but also through regular monitoring of patients during the treatment [28]. It is noteworthy that in our clinic, regular checkups for patients who were receiving oncological therapy have been continuously provided since the beginning of the pandemic, without significant interruptions. Nevertheless, patients’ personal beliefs towards vaccination were also found to be a significant factor in the decision to complete vaccination. Given that many factors may influence the personal attitude towards vaccination, it seems that lack of sufficient information is of utmost importance and results in a negative attitude towards vaccination [29].

Based on the study conducted by Bersanelli et al. [30], it was found that patients with mRCC and metastatic urothelial cancer (mUC) who were treated with immunotherapy had a significantly higher rate of COVID-19 infection. Moreover, a higher rate of permanently discontinued anticancer treatment due to COVID-19 was observed in this patient population. With respect to these findings, it is reasonable for all mRCC patients to receive advice to vaccinate against COVID-19. Furthermore, the recommendations of Associazione Italiana di Oncologia Medica (AIOM) emphasize the importance of providing comprehensive education on vaccine-preventable diseases for cancer patients, including COVID-19 [31]. This is in line with the findings of our study, suggesting that healthcare providers should prioritize educating and guiding cancer patients on vaccination to prevent unnecessary interruptions in cancer treatment which can lead to worse outcomes. Moreover, our study highlights the critical role of doctors involved in cancer patient treatment, particularly urologists in the case of uro-oncology patients, in promoting COVID-19 vaccination acceptance. Furthermore, Martinez-Cannon et al. reported high COVID-19 vaccination rates and positive attitudes among cancer patients in Mexico [32]. They found that unequivocally supporting information about COVID-19 vaccination in mainstream media and building confidence in vaccine safety were associated with higher vaccination rates in cancer patients. Research published by Kelkar et al. on how to improve vaccine acceptance among cancer patients stressed that lack of information, as well as misinformation, considerably impedes vaccine uptake in this patient population [33]. Therefore, they suggest that providing information about vaccine safety from healthcare professionals, particularly treating oncologists, could potentially increase vaccination rates. Hence, effective communication between healthcare providers and cancer patients is crucial for promoting COVID-19 vaccine acceptance in this vulnerable population, as reported by Leak J. et al. [34]. To increase COVID-19 vaccine acceptance among cancer patients, healthcare providers can recommend the vaccine, educate patients on its safety and benefits, address side effect concerns, and provide tailored information based on patients’ medical history and treatment. These interventions can help patients make informed decisions and feel more confident in accepting vaccination against COVID-19.

The limitations of this study are mostly marked by its observational nature. The relatively small number of patients included in our study may be explained by single-center recruitment. Furthermore, a cross-sectional investigation may be of limited value in exploring all factors which may have influenced patients’ decision to vaccinate and is not sufficient in determining the changes in attitude over time. However, these limitations simultaneously reflect the current situation, which is also the quality of this study.

## 5. Conclusions

In conclusion, we found a high rate of COVID-19 vaccination among uro-oncology patients receiving systemic targeted therapy with tyrosine kinase receptor inhibitors (TKIs) for mRCC and systemic therapy with androgen receptor target agents (ARTAs) for mCRPC. Our study showed multiple associations of patients’ sociodemographic characteristics with vaccination rates. Furthermore, we found that consultation with doctors who are particularly involved in oncology treatment and advice received from them are associated with significantly higher vaccination rates among uro-oncology patients. While this study only focused on uro-oncology patients, our findings can be generalized to other patient populations. To increase vaccine acceptance, healthcare professionals should be well informed about vaccines and be prepared to discuss vaccination with their patients.

## Figures and Tables

**Table 1 vaccines-11-00911-t001:** Sociodemographic and clinical characteristics of patients.

Characteristic	Total *n =* 173
Gender, *n* (%)	
Male	149 (86.1)
Female	24 (13.9)
Age, mean ± SD, years	67.4 ± 9.6
Primary uro-oncological disease, *n* (%)	
mRCC	93 (53.8)
mCRPC	80 (46.2)
Duration of disease, median (range), months	51 (4–245)
Duration of systemic treatment, median (range), months	11 (1–102)
Permanent residence, *n* (%)	
In capital city	70 (40.5)
Outside capital city	103 (59.5)
Education, *n* (%)	
Elementary school or less	27 (15.6)
High school	97 (56.1)
Higher education	49 (28.3)
Medical Profession, *n* (%)	
No	170 (98.3)
Yes	3 (1.7)
Number of household members, *n* (%)	
Living alone	25 (14.5)
Living with 1 member	71 (41.0)
Living with ≥2 members	77 (44.5)
Healthcare workers in the household *, *n* (%)	
No	116 (78.4)
Yes	32 (21.6)
Chronic diseases, *n* (%)	
No	62 (35.8)
Yes	111 (64.2)
Comorbidities **, *n* (%)	
Cardiovascular diseases	106 (61.3)
Diabetes mellitus	20 (11.6)
Obesity	2 (1.2)
Chronic obstructive pulmonary disease	1 (0.6)
Chronic renal failure	10 (5.8)

mRCC, metastatic renal cell carcinoma; mCRPC, metastatic castration-resistant prostate cancer; * for patients living with at least one household member; ** patient could choose more than one answer.

**Table 2 vaccines-11-00911-t002:** COVID-19 vaccination and patients’ attitudes related to vaccination.

Characteristic	Total *n =* 173
Vaccination status, *n* (%)	
Unvaccinated	49 (28.3)
Vaccinated	124 (71.7)
Vaccine awareness, *n* (%)	
No	45 (26.0)
Yes	128 (74.0)
Consultation with a doctor, *n* (%)	
No	46 (26.6)
Yes	127 (73.4)
Specialty of doctor who was consulted *, *n* (%)	
Chosen primary care physician	71 (41.0)
Urologist	82 (47.4)
Medical Oncologist	10 (5.8)
Other specialty	8 (4.6)
Doctor’s advice on Vaccination, *n* (%)	
For vaccination	107 (84.3)
Against vaccination	12 (9.4)
Advice not received	7 (5.5)
Both for and against vaccination	1 (0.8)
Vaccination of household members *, *n* (%)	
No	42 (28.4)
Yes	106 (71.6)
Factors that influenced decision to vaccinate **, *n* (%)	
Doctor’s advice	64 (37.0)
Public media	29 (16.8)
Primary malignant disease	63 (36.4)
Family members	21 (12.1)
Personal beliefs	61 (35.3)
I don’t want to answer	8 (4.6)
Previously contracted COVID-19 infection, *n* (%)	
Yes, before systemic treatment started	25 (14.5)
Yes, during systemic treatment	40 (23.1)
No	108 (62.4)
Hesitancy to receive the vaccine ***, *n* (%)	
No	112 (90.3)
Yes	12 (9.7)
Change in attitude towards vaccination ****, *n* (%)	
No	40 (85.1)
Yes	7 (14.9)

* For patients living with at least one household member; ** patient could choose more than one answer; *** for vaccinated patients; **** for unvaccinated patients.

**Table 3 vaccines-11-00911-t003:** The association between the clinical and sociodemographic characteristics of patients and actual vaccination status.

Characteristic	Unvaccinated *n* = 49	Vaccinated *n* = 124	*p*-Value
Gender, *n* (%) Male Female	36 (24.2) 13 (54.2)	113 (75.8) 11 (45.8)	0.002 ^b^
Age, mean ± SD, years	64.1 ± 9.5	68.6 ± 9.4	0.006 ^a^
Primary uro-oncological disease mRCC mCRPC	31 (33.3) 18 (22.5)	62 (66.7) 62 (77.5)	0.115 ^b^
Duration of disease, median (range), months	58 (4–245)	51 (4–212)	0.332 ^c^
Duration of systemic treatment, median (range), months	12 (1–102)	11 (1–96)	0.516 ^c^
Permanent residence, *n* (%) In capital city Outside capital city	16 (22.9) 33 (32.0)	54 (77.1) 70 (68.0)	0.188 ^b^
Education, *n* (%) Elementary school or less High school Higher education	12 (44.4) 32 (33.0) 5 (10.2)	15 (55.6) 65 (67.0) 44 (89.8)	0.002 ^b^
Medical Profession, *n* (%) No Yes	49 (28.8) 0 (0.0)	121 (71.2) 3 (100.0)	0.559 ^d^
Number of household members, *n* (%) Living alone Living with 1 member Living with ≥2 members	8 (32.0) 13 (18.3) 28 (36.4)	17 (68.0) 58 (81.7) 49 (63.6)	0.047 ^b^
Healthcare workers in the household *, *n* (%) No Yes	34 (29.3) 7 (21.9)	82 (70.7) 25 (78.1)	0.405 ^b^
Chronic diseases, *n* (%) No Yes	20 (32.3) 29 (26.1)	42 (67.7) 82 (73.9)	0.391 ^b^
Comorbidities **, *n* (%)			
Cardiovascular diseases	26 (26.4)	78 (73.6)	0.483 ^b^
Diabetes mellitus	8 (40.0)	12 (60.0)	0.218 ^b^
Obesity	1 (50.0)	1 (50.0)	0.483 ^d^
Chronic obstructive pulmonary disease	1 (100.0)	0 (0.0)	0.283 ^d^
Chronic renal failure	4 (40.0)	6 (60.0)	0.472 ^d^

mRCC, metastatic renal cell carcinoma; mCRPC, metastatic castration-resistant prostate cancer; ^a^ Student’s *t*-test, ^b^ χ^2^ test, ^c^ Mann–Whitney U test, ^d^ Fisher’s exact test; * for patients living with at least one household member; ** patient could choose more than one answer.

**Table 4 vaccines-11-00911-t004:** The association between the information received and attitudes towards vaccination in vaccinated and unvaccinated groups of patients.

Characteristic	Unvaccinated *n* = 49	Vaccinated *n* = 124	*p*-Value
Vaccine awareness, *n* (%) No Yes	16 (35.6) 33 (25.8)	29 (64.4) 95 (74.2)	0.211 ^a^
Consultation with doctor, *n* (%) No Yes	23 (50.0) 26 (20.5)	23 (50.0) 101 (79.5)	<0.001 ^a^
Specialty of doctor who was consulted *, *n* (%)			
Chosen primary care physician	19 (26.8)	52 (73.2)	0.703 ^a^
Urologist	9 (11.0)	73 (89.0)	<0.001 ^a^
Medical Oncologist	3 (30.0)	7 (70.0)	1.000 ^b^
Other specialty	2 (25.0)	6 (75.0)	1.000 ^b^
Doctor’s advice on vaccination, *n* (%)			
For vaccination	13 (12.4)	94 (87.9)	
Against vaccination	10 (83.3)	2 (16.7)	
Advice not received	2 (28.6)	5 (71.4)	<0.001 ^b^
Both for and against vaccination	1 (100.0)	0 (0.0)	
Vaccination of household members **, *n* (%) No Yes	35 (83.3) 6 (5.7)	7 (16.7) 100 (94.3)	<0.001 ^a^
Factors which influenced the decision to vaccinate, *n* (%)			
Doctor’s advice	5 (7.8)	59 (92.2)	<0.001 ^a^
Public media	5 (17.2)	24 (82.8)	0.147 ^a^
Primary malignant disease	17 (27.0)	46 (73.0)	0.767 ^a^
Family members	1 (4.8)	20 (95.2)	0.011 ^a^
Personal beliefs	24 (39.3)	37 (60.7)	0.018 ^a^
Previously contracted COVID-19 infection, *n* (%)			
Yes, before systemic treatment started	13 (52.0)	12 (48.0)	
Yes, during systemic treatment	12 (30.0)	28 (70.0)	0.011 ^a^
No	24 (22.2)	84 (77.8)	

^a^ χ^2^ test, ^b^ Fisher’s exact test; * patient could choose more than one answer; ** for patients living with at least one household member.

## Data Availability

All data shown in this study are included in this published article.

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
