# Peer review of "Role of Healthcare Professionals and Sociodemographic Characteristics in COVID-19 Vaccination Acceptance among Uro-Oncology Patients: A Cross-Sectional Observational Study"

_vaccines, 2023, doi:10.3390/vaccines11050911_

Round 1

Reviewer 1 Report

Dear Author,

Greetings1

1Role of Healthcare Professionals and Sociodemographic Char- 2

acteristics in COVID-19 Vaccination Acceptance among Uro- 3

Oncology Patients: A Cross Sectional Observational Study(Title is too long)

2This was a single-center cross-sectional observational study conducted in 14

March 2022. We collected information on patients' sociodemographic characteristics, vaccination 15

status, awareness and attitudes about COVID-19 vaccination(Please revise)

3We noticed higher vaccination rates in male patients 18

(p=0.002), older (p=0.006), highly educated (p=0.002), and patients who lived with one household 19

member (p=0.047).(male age and education its not clear)

4Since the first case of SARS-CoV-2 infection was reported in Wuhan, China in December 312019, the COVID-19 pandemic has rapidly spread,((Introduction should not start with since)

5 table and figures should be as per journal format

6 Taguchi analysis of statistics will work 

7 REFERENCES  2,5 23 ARE NOT Arranged aS PER FORMAT

Please include questionnaire as per the    clinical and sociodemographic characteristics of patients

BEST 

Minor editing of English language required

Author Response

Dear Reviewer,

Many thanks for taking the time to review our manuscript and for providing constructive feedback. We have addressed the comment and included changes/additional data, which have improved our manuscript. The response to the comment is below.

  1. Role of Healthcare Professionals and Sociodemographic Characteristics in COVID-19 Vaccination Acceptance among Uro-Oncology Patients: A Cross Sectional Observational Study (Title is too long)

We thank the reviewer for this comment. We agree that title is long. However, this title in the best way accentuate the main findings in our study. If reviewer agree, we would like to keep the title unchanged. Yet, if changes are required, then we suggest this title:

"Role of Healthcare Professionals and Sociodemographic Characteristics in COVID-19 Vaccination Acceptance among Uro-Oncology Patients"

  1. This was a single-center cross-sectional observational study conducted in March 2022. We collected information on patients' sociodemographic characteristics, vaccination status, awareness and attitudes about COVID-19 vaccination (Please revise)

  1. We noticed higher vaccination rates in male patients (p=0.002), older (p=0.006), highly educated (p=0.002), and patients who lived with one household member (p=0.047) (male age and education its not clear)

We thank the reviewer for these comments. To comply with the both of the reviewer comments, we have revised the whole abstract. Revised text corresponding to reviewer suggestions is:

"Data on patients' sociodemographic characteristics, vaccination status, awareness and attitudes about COVID-19 vaccination, were collected from questionnaires completed by the patients. A total of 173 patients were enrolled in this study, and 124 (71.7%) of them completed the COVID-19 vaccination. Significantly higher vaccination rates were found in male patients, and also in older patients, highly educated patients, and those who lived with one household member."

  1. Since the first case of SARS-CoV-2 infection was reported in Wuhan, China in December 312019, the COVID-19 pandemic has rapidly spread (Introduction should not start with since)

We thank the reviewer for this comment. To comply with the reviewer comment, we have made a changes in this first sentence of Introduction section. Please, find it in revised version of the manuscript.

"Soon after the first case of SARS-CoV-2 infection was reported in Wuhan, China in December 2019, the COVID-19 pandemic has rapidly spread, causing significant changes in all areas of life across the globe [1]."

  1. Table and figures should be as per journal format

We thank the reviewer for this comment. Please, find all corrections that we made in Tables in revised version of the manuscript.

  1. Taguchi analysis of statistics will work 

We thank the reviewer for this comment. We agree that Taguchi method which identifies the significant level of a factor which affects the specific performance parameter, may add value to our work. However, we were concerned that the amount of data we obtained was not sufficient to have enough for this type of statistics. Besides, none of the authors were familiar with Taguchi method and how to interpret the results which are only relative and do not exactly indicate what parameter has the highest effect on the performance characteristic value. Therefore, we did not use this type of statistics in our study.

  1. References 2,5 and 23 are not arranged as per format

We thank the reviewer for this comment. Please find corrected references in revised version of the manuscript.

  1. Please include questionnaire as per the clinical and sociodemographic characteristics of patients

We thank the reviewer for this comment. The English version of questionnaire which was used in this study was already submitted as Suplementary File together with Manuscript File.

It is .rar file (compressed archive) and therfore problems with extracting the files might occur.

Data on clinical characteristics, underlying oncological disease, and systemic therapy were obtained from patients' electronic and paper medical records. We collected information on patients' sociodemographic characteristics, vaccination status, awareness and attitudes about COVID-19 vaccination, and factors that influenced their decisions from questionnaires completed by the patients. Please, find the questionnaire in Manuscript Information Overview Suplementary File link at Journal web portal.

Reviewer 2 Report

Article to be published in its current form. Please adapt the bibliography to the requirements of the journal.

Author Response

Dear Reviewer,

Many thanks for taking the time to review our manuscript and for providing constructive feedback. We have addressed the comment and included changes/additional data, which have improved our manuscript. The response to the comment is below.

  1. Please adapt the bibliography to the requirements of the journal.

We thank the reviewer for this comment. Please find corrected references in revised version of the manuscript.

Reviewer 3 Report

Please improve the discussion:

- what type of interventions might implement vaccine acceptance?

- why were the patients in immunotherapy not considered?

- add and discuss PMID: 34707691; https://doi.org/10.1016/j.esmoop.2023.101215; PMID: 36913048; doi: 10.3390/healthcare9030351; PMID: 34137034

NA

Author Response

Dear Reviewer,

Many thanks for taking the time to review our manuscript and for providing constructive feedback. We have addressed the comment and included changes/additional data, which have improved our manuscript. The response to the comment is below.

  1. What type of interventions might implement vaccine acceptance?

  1. Add and discuss - PMID: 34707691;https://doi.org/10.1016/j.esmoop.2023.101215; 

PMID: 36913048; doi: 10.3390/healthcare9030351; PMID: 34137034

We thank the reviewer for these comments. We comply with the reviewer comment and we added more discussion and literature review, simultaneously responding to the comment No1 as well. Therefore we added the following text in Discussion section:

"Based on the study conducted by Bersanelli et al. [30], it was found that patients with mRCC and metastatic urothelial cancer (mUC) who were treated with immunotherapy, had a significantly higher rate of COVID - 19 infection. Moreover, a higher rate of permanently discontinued anticancer treatment due to COVID - 19 was observed in this patient population. With respect to these findings, it is reasonable for all mRCC patients to receive advise to vaccinate against COVID-19. Furthermore, the recommendations of AIOM (Associazione Italiana di Oncologia Medica) emphasize the importance of providing comprehensive education on vaccine-preventable diseases for cancer patients, including COVID-19 [31]. This is in line with the findings of our study, suggesting that healthcare providers should prioritize educating and guiding cancer patients on vaccination to prevent unnecessary interruptions in cancer treatment which can lead to worse outcomes. Moreover, our study highlights the critical role of doctors involved in cancer patient treatment, particularly urologists in the case of uro-oncology patients, in promoting COVID-19 vaccination acceptance. Furthermore, Martinez-Cannon et al. reported high COVID-19 vaccination rates and positive attitudes among cancer patients in Mexico [32]. They found that unequivocally supporting information about COVID-19 vaccination in mainstream media and building confidence in vaccine safety were associated with higher vaccination rates in cancer patients. Research published by Kelkar et al. on how to improve vaccine acceptance among cancer patients, stressed that lack of information, but also misinformation considerably impede vaccine uptake in this patients population [33]. Therefore, thay suggest that providing information about vaccine safety from healthcare professionals, particularly treating oncologists, could potentially increase vaccination rates. Hence, effective communication between healthcare providers and cancer patients is crucial for promoting COVID-19 vaccine acceptance in this vulnerable population, reported by Leak J. et al [34]. To increase COVID-19 vaccine acceptance among cancer patients, healthcare providers can recommend the vaccine, educate patients on its safety and benefits, address side effect concerns, and provide tailored information based on patients' medical history and treatment. These interventions can help patients make informed decisions and feel more confident in accepting vaccination against COVID-19."

  1. Why were the patients in immunotherapy not considered?

We thank the reviewer for this comment. At the time when this study was conducted, immunotherapy was not yet available in Serbia. Therefore, it was not possible to include patients receiving immunotherapy in the study.

Reviewer 4 Report

Thanks for the invitation. This study focuses on one of the ongoing hot topics, but some modifications need to be made. Regression can be added.

Other comments are:

    1.  the sample size of 120 is too small to get conclusive results.
  1. For analyzing participant's data from online database, several factors are always difficult to overcome, including data missing, poor qualification, participation duplication and so on.
  2. please add the strength and limitations of this study
  3. In the methods, mention it is an online questionnaire survey. Mention the validity and reliability of the questionnaire and how it was measured.
  4. Which social media platform was used to distribute the questionnaire?

Author Response

Dear Reviewer,

Many thanks for taking the time to review our manuscript and for providing constructive feedback. We have addressed the comment and included changes/additional data, which have improved our manuscript. The response to the comment is below.

  1. Regression can be added.

We thank the reviewer for this comment. Given that some of our examined variables are in obvious mutual connection (such as consultation with a doctor regarding vaccination, consultation with a specialist regarding vaccination, advice from a doctor on vaccination, etc.), we believe that the obtained findings would not be highlighted by regression analysis.

  1. The sample size of 120 is too small to get conclusive results.

We thank the reviewer for this comment. A total of 173 patients were included in our study, of which 124 were vaccinated against COVID 19. Unfortunately, in our country, and therefore in our institution, specific oncological therapy for mCRCP and mRCC is received by a relatively small number of patients.

  1. For analyzing participant's data from online database, several factors are always difficult to overcome, including data missing, poor qualification, participation duplication and so on.

We thank the reviewer for this comment. As we stated in the Material and Methods section, the patient survey was not conducted online, but directly during routine clinical visits of patients at th Clinic of Urology in March 2022.

  1. Please add the strength and limitations of this study

We thank the reviewer for this comment. At the end of the Discussion section, we pointed out the limitations of our study. Our study is an observational study, conducted on a relatively small number of patients and not all factors that could potentially affect vaccination were included. The importance of our results is highlighted by the fact that they represent the current situation regarding vaccination among uro-oncology patients.

  1. In the methods, mention it is an online questionnaire survey. Mention the validity and reliability of the questionnaire and how it was measured.

We thank the reviewer for this comment. Our research was conducted by directly surveying patients during their regular visits to the Clinic. The original questionnaire constructed at the Clinic for Urology which was used for the purposes of this research is based on questionnaires from similar studies.

  1. Which social media platform was used to distribute the questionnaire?

We thank the reviewer for this comment. We did not use social media platforms to survey the patients, but we surveyed the patients in the manner indicated in the answers to question No. 3 and No. 5.